# Parallel Training of GRU Networks with a Multi-Grid Solver for Long Sequences

**Gordon Euhyun Moon**
Korea Aerospace University
`ehmoon@kau.ac.kr`

**Eric C. Cyr**
Sandia National Laboratories
`eccyr@sandia.gov`

## Abstract

Parallelizing Gated Recurrent Unit (GRU) networks is a challenging task, as the training procedure of GRU is inherently sequential. Prior efforts to parallelize GRU have largely focused on conventional parallelization strategies such as data-parallel and model-parallel training algorithms. However, when the given sequences are very long, existing approaches are still inevitably performance limited in terms of training time. In this paper, we present a novel parallel training scheme (called parallel-in-time) for GRU based on a multigrid reduction in time (MGRIT) solver. MGRIT partitions a sequence into multiple shorter sub-sequences and trains the sub-sequences on different processors in parallel. The key to achieving speedup is a hierarchical correction of the hidden state to accelerate end-to-end communication in both the forward and backward propagation phases of gradient descent. Experimental results on the HMDB51 dataset, where each video is an image sequence, demonstrate that the new parallel training scheme achieves up to $6.5\times$ speedup over a serial approach. As efficiency of our new parallelization strategy is associated with the sequence length, our parallel GRU algorithm achieves significant performance improvement as the sequence length increases.

## 1 Introduction

Recently, the model complexity of Deep Neural Networks (DNN) has grown to keep pace with the increasing scale of data. Hence, parallelization of DNNs is becoming important to achieve high quality models in tractable computational run times (Ben-Nun & Hoefler, 2019). Sequence modeling is one learning task that suffers from substantial computational time due to data size. The goal is to capture temporal dependencies within a sequence. Several different deep learning approaches addressing sequence tasks have been proposed, including 3D-CNNs (Carreira & Zisserman, 2017), LSTM (Srivastava et al., 2015), and Transformers (Girdhar et al., 2019). Focusing on sequence classification, this paper proposes a parallelization strategy for a gated recurrent unit (GRU) which is a special type of recurrent neural network (RNN) architecture.

Parallel algorithms for training neural networks are generally broken into two classes; *data-parallelism* distributes a batch across processors, and *model-parallelism* distributes the architecture (Ben-Nun & Hoefler, 2019). Data-parallelism inherent in mini-batch stochastic gradient descent (SGD) is the primary strategy for parallelizing RNN architecture (You et al., 2019; Huang et al., 2013). However, the accuracy of mini-batch SGD degrades as the number of sequences in each mini-batch increases. Moreover, although multiple mini-batches are processed in parallel, the training procedure of each sequence (within a mini-batch) is invariably sequential. Alternatively, the RNN model can be parallelized across hidden layers (Wu et al., 2016; MXNet, 2018). This model-parallelism approach is beneficial when the network requires a large number of layers. However, even large-scale RNN such as the Google Neural Machine Translation (GNMT) developed by Wu et al. (2016) have only 8 LSTM layers. Therefore, the scalability of layer-based model-parallelism is limited. A common problem in data-parallel and model-parallel algorithms for GRU is that the training execution time increases with the sequence length. At the heart of this difficulty is a fundamental need to process sequences in a serial fashion, yielding forward and backward propagation algorithms that are inherently sequential, and thus hard to parallelize.

We propose a parallel-in-time (PinT) training method for GRU networks with long sequences. To achieve this, we adapt a multigrid reduction in time (MGRIT) solver for forward and backward propagation. Within the numerical methods community, a resurgence in PinT has paralleled increasing computational resources (Gander, 2015; Ong & Schroder, 2020). These efforts have even been applied to the training of neural Ordinary Differential Equations (ODEs) (Schroder, 2017; Gunther et al., 2020; Kirby et al., 2020; Cyr et al., 2019), where inexact forward and back propagation is exploited to achieve parallel speedups. To date, these techniques have not been applied to RNN.

Following Jordan et al. (2021), GRU networks can be written as an ODE to facilitate application of MGRIT. Different from existing parallel training algorithms for RNN, our MGRIT parallel training scheme partitions a sequence into multiple shorter sub-sequences and distributes each sub-sequence to different processors. By itself, this provides local improvement of the hidden states, yet global errors remain. To correct these errors, propagation is computed on a coarse representation of the input sequence requiring less computational work while still providing an improvement to the original hidden states. This process is iterated to the accuracy required for training. Applying this algorithm to the classic GRU networks will achieve parallelism. However, this training algorithm will not lead to accurate networks due to stability problems on the coarse grid. This emphasizes the challenges of choosing a proper coarse grid model for neural networks, and multigrid algorithms in general. To alleviate this, we develop a new GRU architecture, or discretization of the ODE, which we refer to as *Implicit GRU* that handles stiff modes in the ODE. This is required for application of the MGRIT algorithm where coarse sub-sequences providing the correction, correspond to discretizations with more strict numerical stability restrictions. We also compare the accuracy of serial versus parallel inference. This ensures that the network is not compensating for the error introduced by the MGRIT procedure, and is a study that has not been considered in prior work.

Two datasets are chosen for evaluation of the PinT algorithm. The first is the UCI-HAR dataset for human activity recognition using smartphones (Anguita et al., 2013). This small dataset is used to demonstrate the algorithm can obtain similar accuracy to classical training approaches, while also demonstrating speedups. The second learning problem uses the HMDB51 dataset for human activity classification in videos (Kuehne et al., 2011). As the size of each image in the video is too large to directly use it as an input for a RNN-based model, we use a pre-trained CNN model to generate low-dimensional input features for the GRU network. This approach captures spatio-temporal information in videos (Yue-Hei Ng et al., 2015; Donahue et al., 2015; Wang et al., 2016). In our experiments, extracted image features from the ResNet pre-trained on ImageNet are used as input sequences for GRU networks. Using this architecture on the HMDB51 dataset, the MGRIT algorithm demonstrates a $6.5\times$ speedup in training over the serial approach while maintaining accuracy.

The paper is organized as follows. Section 2 presents background on GRUs and their formulation as ODEs. Section 3 details PinT training for GRU using the MGRIT method. Section 4 compares the performance of the proposed approach on sequence classification tasks. We conclude in Section 5. The appendix provides greater detail on the MGRIT method including a serial implementation and parameters used in the experiments.

## 2 GATED RECURRENT UNIT

The GRU network, proposed by Cho et al. (2014), is a type of RNN and a state-of-the-art sequential model widely used in applications such as natural language processing and video classification. Each GRU cell at time $t$ takes input data $x_t$ and the previous hidden state $h_{t-1}$, and computes a new hidden state $h_t$. The cell resets or updates information carefully regulated by gating units in response to prior hidden state and input sequence data stimulation as follows:

$$
\begin{aligned}
r_t &= \sigma(W_{ir}x_t + b_{ir} + W_{hr}h_{t-1} + b_{hr}) \\
z_t &= \sigma(W_{iz}x_t + b_{iz} + W_{hz}h_{t-1} + b_{hz}) \\
n_t &= \varphi(W_{in}x_t + b_{in} + r_t \odot (W_{hn}h_{t-1} + b_{hn})) \\
h_t &= z_t \odot h_{t-1} + (1 - z_t) \odot n_t.
\end{aligned}
\tag{1}
$$

We refer to this method as *Classic GRU* in the text.

**GRUs to ODEs** Recent work has shown how RNN can be modified to obtain an ODE description (Chang et al., 2019; Habiba & Pearlmutter, 2020). Of particular interest here is those works for

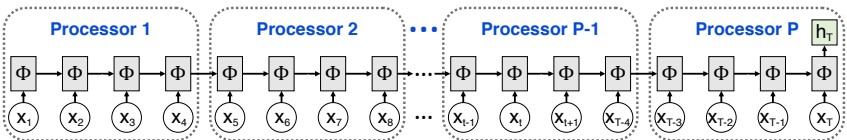

Figure 1: Parallel distribution of work in parallel-in-time training.

GRU networks (De Brouwer et al., 2019; Jordan et al., 2021) where the ODE is

$$\partial_t h_t = \underbrace{-(1 - z_t) \odot h_t}_{\text{stiff mode}} + (1 - z_t) \odot n_t. \tag{2}$$

Jordan et al. (2021) present a detailed description of the network behavior. For our purposes, note that the first term can generate stiff dynamics that drive the hidden state $h$ towards a manifold specified by the data. The second term injects changes from the data. The term $z_t$ controls the decay rate towards the manifold for each component of the hidden state. If $z_t$ is near 1, then the rate is slow. If $z_t$ is close to zero the decay rate is rapid and the first term can be stiff, constraining the choice of time step (for numerical treatment of stiff ODE's see Wanner & Hairer (1996)).

The classic GRU method in Eq. 1 is rederived using an explicit Euler scheme with a unit time step to integrate Eq. 2. Adding and subtracting $h_{t-1}$ on the right hand side of the hidden state update, and for notational purposes multiplying by $\gamma = 1$, yields

$$h_t = \Phi_\gamma(x_t, h_{t-1}) := h_{t-1} + \gamma \left( -(1 - z_t) \odot h_{t-1} + (1 - z_t) \odot n_t \right). \tag{3}$$

The notation $\Phi_\gamma$, introduced here, denotes the classic GRU hidden state update if $\gamma = 1$.

## 3 PARALLEL-IN-TIME TRAINING

Training GRU networks is typically done with the backpropagation through time (BPTT) algorithm (Sutskever, 2013). BPTT performs a forward evaluation of the neural network, followed by backpropagation to compute the gradient. Both forward and backward propagation are computationally expensive algorithms that do not naturally lend themselves to parallelization in the time domain. Despite this, we develop a methodology achieving the parallel distribution shown in Fig. 1. This distribution is enabled by computing the forward and backward propagation inexactly, trading inexactness for parallelism and ultimately run time speedups. MGRIT, a multigrid method, is used to compute the hidden states for both forward and backward propagation.

Multigrid methods are used in the iterative solution of partial differential equations (Brandt, 1977; Douglas, 1996; Briggs et al., 2000; Trottenberg et al., 2000). Recent multigrid advances have yielded the MGRIT method achieving parallel-in-time evolution of ODEs like Eq. 2 (see for instance Falgout et al. (2014); Dobrev et al. (2017); Gunther et al. (2020)). Thus, multigrid and MGRIT methods are reasonably well developed; however, they remain quite complex. Here, we briefly outline the algorithm for forward propagation, and present a brief motivation for the method in Appendix A.1. In Section 3.3, an implicit formulation for GRU that handles the stiff modes is proposed. This is the key advance that enables the use of MGRIT for training GRU networks.

### 3.1 NOTATION AND PRELIMINARIES

Denote a sequence by $\vec{u} = \{u_t\}_{t=0}^T$ where the length $T$ will be clear from context, and $u_0 = 0$ by assumption. This notation is used for the input data sequence $\vec{x}$ and the sequence of hidden states $\vec{h}$. The GRU forward propagator in Eq. 3 can be written as the system of nonlinear algebraic equations

$$f(\vec{h}) = \{h_t - \Phi_1(x_t, h_{t-1})\}_{t=1}^T. \tag{4}$$

When $f(\vec{h}) = 0$, then $\vec{h}$ is the hidden state for the GRU networks with input sequence $\vec{x}$.

A two-level multigrid algorithm uses a *fine* sequence on level $l = 0$, corresponding to a data input of length $T$. The *coarse* level, defined on level $l = 1$, has data of length $T/c_f$ where $c_f$ is an integer coarsening factor. Sequences are transferred from the fine to coarse level using a *restriction* operator

$$\vec{U} = \mathcal{R}(\vec{u}) := \{u_{c_f t}\}_{t=1}^{T/c_f}. \tag{5}$$

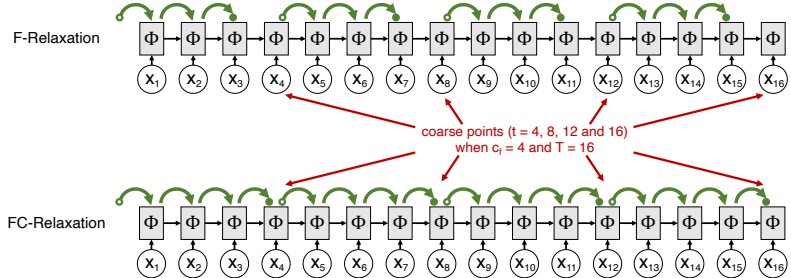

Figure 2: For $c_f = 4$ with $T = 16$, this image shows $F$- and $FC$-relaxation. In both cases, forward propagation is performed serially starting at the open green circle and ending at the closed one. Each propagation step is performed in parallel, with four-way parallelism depicted in the image.

Applying $\mathcal{R}$ to $\vec{x}$ gives a coarse sub-sequence of the original. Time points on the fine level that are included in the coarse level are referred to as *coarse points* (see Fig. 2). Transfers from coarse to fine grids are denoted by the *prolongation* operator

$$\vec{u} = \mathcal{P}(\vec{U}) := \{U_{\lfloor t/c_f \rfloor}\}_{t=1}^{T}. \tag{6}$$

Prolongation fills intermediate values using replication of the prior coarse point. The nonlinear system from Eq. 4 is modified using $\Delta t = c_f$, yielding the residual for the coarse network:

$$f_c(\vec{H}) = \left\{ H_t - \Phi_{c_f}(X_t, H_{t-1}) \right\}_{t=1}^{T/c_f}. \tag{7}$$

Notice that in addition to a larger time step, the length of sequence $\vec{H}$ is smaller than the fine grid's.

Finally, we introduce the notion of *relaxation* of the hidden state. Here, we define this formally, while in Fig. 2 a schematic view of these algorithms is taken. Relaxation applies propagation to an initial condition for the hidden state in a parallel fashion. $F$-*relaxation* is defined in Algorithm 1 as $\Psi_F(c_f, \vec{x}, \vec{h})$ and demonstrated in the top of Fig. 2. $F$-relaxation uses the hidden state at the coarse points, and fills in the fine points up to the next coarse point (exclusive). $FC$-relaxation, $\Psi_{FC}$, is identical to $F$-relaxation except the next coarse point is included (the bottom row in Fig. 2, where closed circles fall on the coarse grid point). Combining $F$- and $FC$-relaxation yields $FCF$-*relaxation* used in MGRIT to ensure sequence length independent convergence

$$\vec{h}' = \Psi_{FCF}(c_f, \vec{x}, \vec{h}) := \Psi_F(c_f, \vec{x}, \Psi_{FC}(c_f, \vec{x}, \vec{h})). \tag{8}$$

The available parallelism for all relaxation methods is equal to the number of coarse points.

---

**Algorithm 1:** $\Psi_F(\gamma, \vec{x}, \vec{h})$ - $F$-relaxation

1. **parallel_for** $t_c \leftarrow 1$ **to** $T/\gamma$
2. $\quad t = \gamma t_c$
3. $\quad h'_t = h_t$
4. $\quad$ **for** $\tau \leftarrow t$ **to** $t + \gamma - 1$
5. $\quad\quad h'_{\tau+1} = \Phi_\gamma(x_\tau, h'_\tau)$
6. **return** $\vec{h}'$

---

### 3.2 Multi-grid in Time

Algorithm 2 presents pseudocode for the two-level MGRIT method for forward propagation. A related schematic is presented in Fig. 3 where line numbers are included in parentheses. In Fig. 3, the fine grid on the top corresponds to an input sequence of length 16. It is depicted as being distributed across 4 processors. The coarse grid representation is a sub-sequence of length 4, its propagation is limited to a single processor. Red arrows in Fig. 3 indicate grid transfers, while green arrows indicate relaxation and propagation through the network. Appendix A.1 provides a brief motivation for MGRIT, and a serial implementation is included in the supplementary materials.

---

**Algorithm 2:** MGProp$(\vec{x}, \vec{h})$ - Forward propagation using multigrid

// Relax the initial guess
1. $\vec{h}' = \Psi_{FCF}(\vec{x}, \vec{h})$
// Transfer approximate hidden state to the coarse grid
2. $\vec{r}_c = \mathcal{R}(f(\vec{h}'))$
3. $\vec{h}'_c = \mathcal{R}(\vec{h}')$
// Compute coarse grid correction, return to fine grid
4. Solve for $\vec{h}^*_c$: $f_c(\vec{h}^*_c) = f_c(\vec{h}'_c) - \vec{r}_c$
5. $h'' = h' + \mathcal{P}(h^*_c - h'_c)$
// Update the fine grid solution, and relax on fine level
6. **return** $\Psi_F(\vec{x}, \vec{h}'')$

---

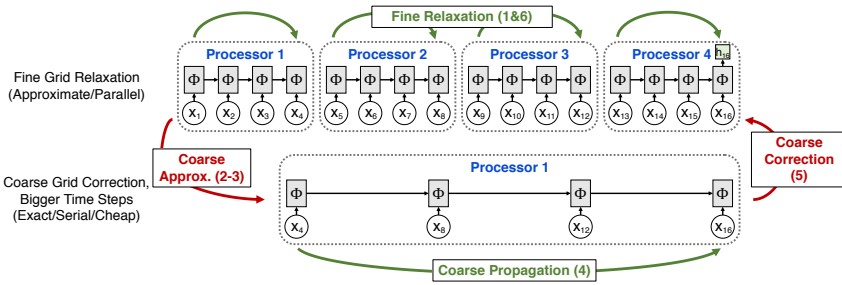

Figure 3: The two level MGRIT method. Green arrows represent relaxation, and forward propagation. Grid transfers are represented by red arrows. Numbers in parentheses are lines in Algorithm 2.

Line 1 in Algorithm 2 relaxes the initial guess, and improves the fine grid hidden state. $FCF$-relaxation is typically used to obtain sequence length independent convergence, though often on the fine level $F$-relaxation is used. Regardless, relaxation is applied in parallel yielding a processor local improved hidden state. The fine level residual given the relaxed hidden state is

$$\vec{r} = f(\vec{h}'). \tag{9}$$

Let $\vec{h}^*$ be the forward propagated hidden state that satisfies $f(\vec{h}^*) = 0$. This is easily computed by serial forward propagation. Rewriting $\vec{r}$ in terms of $\vec{h}'$ and $\vec{h}^*$ gives

$$\vec{r} = f(\vec{h}') - f(\vec{h}^*). \tag{10}$$

Coarsening the residual using the restriction operator from Eq. 5, we arrive at an equation for the coarse grid approximation of $\vec{h}^*$

$$f_c(\vec{h}^*_c) = f_c(\mathcal{R}(\vec{h}')) - \mathcal{R}(\vec{r}). \tag{11}$$

The coarse grid approximation is computed using Eq. 11 in Line 4 of the algorithm. This computation has reduced run time relative to the fine grid because the input sequence is shorter. Line 5 transfers a coarse approximation to the fine grid. The final step in the algorithm is $F$-relaxation on Line 6 that propagates the corrected solution from the coarse points to the fine points.

**Remark 1** (Multiple Iterations). *MGRIT is an iterative algorithm. One pass through Algorithm 2 may not provide sufficient accuracy in the hidden state for use in neural network training. However, in Gunther et al. (2020), relatively few iterations were needed for successful training.*

**Remark 2** (Beyond Two-Level). *As presented, this is a two level algorithm. The coarse grid correction in line 4 provides an end-to-end correction at a reduced run-time of $T/c_f$. However, for large $T$, this may be prohibitively expensive. The solution is to recognize that the coarse grid problem itself can be solved using MGRIT in a recursive fashion. Thus, new levels are added to reduce the run time of the coarse problem. The run time of level $l$ with $T$ time steps scales as $c(l) = T/P + c(l+1)$, where $P$ is the number of processors. Therefore, the total run time for $L$ levels scales as*

$$c(0) = \sum_{l=0}^{L-1} \frac{T}{Pc_f^l} \leq \frac{T}{P} \sum_{l=0}^{\infty} \frac{1}{c_f^l} \leq \frac{T}{P} \frac{c_f}{c_f - 1}. \tag{12}$$

### 3.3 COARSE GRID PROPAGATORS

The discussion above assumes that all levels use the classic GRU propagator in Eq. 3 (forward Euler with a time step of $\Delta t = 1$). This presents a problem for coarsening. Consider only the stiff term in the dynamics from Eq. 2 and assume that $z \in [0, 1]$ is constant for all time. This results in the model ODE $\partial_t h = -(1 - z) \odot h$. Discretizing with forward Euler, stable time steps satisfy

$$|1 - \Delta t(1 - z)| \leq 1 \Rightarrow 0 \leq \Delta t \leq 2(1 - z)^{-1}. \tag{13}$$

The model ODE shares the stiff character of the GRU formulation, and the time step is restricted to be $\Delta t < 2$ assuming the worst case $z = 0$ ($z = 1$ implies there is no time step restriction). On the fine grid with $\Delta t = 1$, forward Euler is not unstable. However, on coarse grids, where the time step is equal to $c_f^l$, stability of forward and back propagation is a concern. In practice, we found instabilities on coarse grids materialize as poor training, and generalization.

To alleviate problems with stiffness, the traditional approach is to apply an implicit method. For instance, applying a backward Euler discretization to the model ODE yields a stability region of

$$|1 + \Delta t(1 - z)|^{-1} \leq 1. \tag{14}$$

Hence, the time step is unlimited for the model problem (for $1 - z < 0$ the ODE is unstable).

A backward Euler discretization of Eq. 2 requiring a nonlinear solver is possible; however, it is likely needlessly expensive. We hypothesize, considering the model problem, that the stiffness arises from the first term in Eq. 2. The second term incorporates the input sequence data, changing the hidden state relatively slowly to aggregate many terms over time. Finally, the nonlinearity in $z$ may be problematic; however, this term primarily controls the decay rate of the hidden state.

As a result of this discussion, we propose an implicit scheme to handle the term labeled "stiff mode" in Eq. 2. The discretization we use computes the updated hidden state to satisfy

$$(1 + \Delta t(1 - z_t)) \odot h_t = h_{t-1} + \Delta t(1 - z_t) \odot n_t. \tag{15}$$

We refer to this method as *Implicit GRU*. Here, the term $(1 - z_t)$ on the left hand side and the right hand side are identical and evaluated at the current data point $x_t$ and the old hidden state $h_{t-1}$. Compared to classic GRU, this method inverts the left-hand side operator to compute the new hidden state $h_t$. This is easily achieved because $(1 + \Delta t(1 - z_t))$ is a diagonal matrix. Thus, the evaluation cost of Eq. 15 is not substantially different from Eq. 3. Yet the (linear) stiffness implied by the first term in the ODE is handled by the implicit time integration scheme. To simplify notation, Eq. 15 is rewritten by defining an update operator $\Phi_\gamma$ (similar to Eq. 3)

$$h_t = \Phi_\gamma(x_t, h_{t-1}; ) := (1 + \gamma(1 - z_t))^{-1} \odot (h_{t-1} + \gamma(1 - z_t) \odot n_t). \tag{16}$$

The subscript $\gamma$ is the time step size, and the learned parameters are identical to classic GRU. This propagator replaces the classic GRU propagator in all levels of the MGRIT algorithm.

### 3.4 TRAINING WITH MGRIT

Training using MGRIT follows the traditional approaches outlined in Goodfellow et al. (2016). In fact, the MGRIT algorithm has been implemented in PyTorch (Paszke et al., 2019), and the Adam implementation is used without modification. The difference comes in the forward propagation, and backward propagation steps. These are replaced with the MGRIT forward and backward propagation. We provide a more detailed description of this approach in Appendix A.2. However, the iterations used for these steps are much fewer than would be required to achieve zero error in the propagation. We found that using 2 MGRIT iterations for the forward propagation, and 1 iteration for back propagation was all that is required to reduce the training loss.

**Remark 3** (Serial vs. Parallel Inference). *As training with MGRIT is inexact, a question arises: does inference with the network evaluated in parallel differ from when the network is evaluated in serial? Our results below indicate consistency between serial and parallel inference even when training is done with MGRIT. This suggests that error introduced by MGRIT is not a severe limitation.*

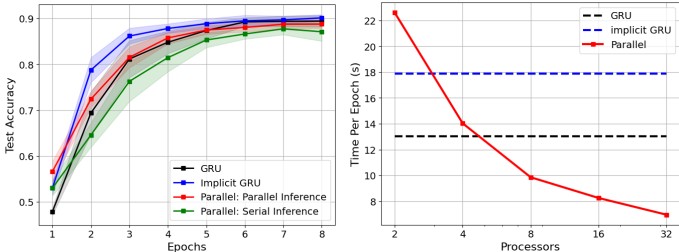

Figure 4: For the UCI-HAR dataset: (left) Accuracy for classic, implicit and parallel training of a GRU network on the UCI-HAR dataset. (right) Run time per epoch for training.

## 4 EXPERIMENTAL EVALUATION

In this section, we evaluate the performance of our parallel-in-time training approach on sequence classification tasks such as human activity recognition and video classification. Experiments are run on a multi-node supercomputer, named Attaway at Sandia National Laboratories, with each node comprised of 2 sockets with a 2.3 GHz Intel Xeon processor, each processor has 18 cores yielding 36 cores per node. Inter-process communication is handled using the Message Passing Interface (MPI) (Gropp et al., 1999) which has been optimized for the Intel Omni-Path interconnect. For the video classification dataset, 9 OpenMP threads were used per MPI rank, yielding 4 ranks per node.

We used two publicly available datasets – UCI-HAR (Anguita et al., 2013) and HMDB51 (Kuehne et al., 2011). The UCI-HAR dataset is a collection of 10299 sequences of human activities labeled with 6 activities such as walking, walking upstairs, walking downstairs, sitting, standing, and laying. Each sequence consists of 128 consecutive input data samples of size 9 measuring three different types of motion recorded by smartphone accelerometers. The HMDB51 dataset contains 6726 video clips recording 51 human activities. To limit computational costs, in several of our experiments a subset of classes are used (see Table 2 in the appendix). For consistency, each video clip is converted (truncated/padded) into 128 frames. Each image in the sequence is of size $240 \times 240$ pixels, and is passed through ResNet18, 34 or 50 networks pre-trained on ImageNet to extract features for the GRU networks (Deng et al., 2009; He et al., 2016). See Appendix A.3.2 for architectural details.

The software implementation of the MGRIT GRU propagation algorithm is a combination of the C++ library XBraid and PyTorch. The combined capability is referred to as TorchBraid[1]. The implementation leverages performance enhancements developed for classic GRU within PyTorch. While *Implicit GRU* has been carefully implemented, further optimizations are certainly possible to achieve greater computational performance. A GPU implementation of the MGRIT algorithm is under development. Finally, it is important to note that the time parallel implementation does not interfere substantially with traditional data and model forms of parallelism.

### 4.1 UCI-HAR DATASET

In Fig. 4, the left image shows the test accuracy variability over 16 training runs for the UCI-HAR dataset. The set of parameters is detailed in Table 1 of the appendix. Solid lines are the mean accuracy, while the colored shadows indicate one standard deviation. The classic GRU (black) achieves 90% accuracy. Implicit GRU (blue line) does as well though with a more rapid increase with epochs. When training with MGRIT, three levels with a coarsening rate of 4 are used for each sequence of length 128. This yields a hierarchy of grids of size 128, 32, and 8. Here, we see that the test accuracy using parallel inference (red line) is modestly lower than the serial case. Inference with the serial algorithm suffers a little more. The plot on the right demonstrates a roughly $2\times$ speedup over the classic GRU method in serial (black dashed line) when using the MGRIT algorithm. This dataset is very small, and likely suffers from the effects of Amdahls' law (see Sun & Ni (1993) for discussions of Amdahl's law). Thus, even achieving $2\times$ speedup is notable. Beyond 32 MPI ranks there is no further speedup due to communication swamping potential gains from parallelism.

---

[1]Available at https://github.com/Multilevel-NN/torchbraid

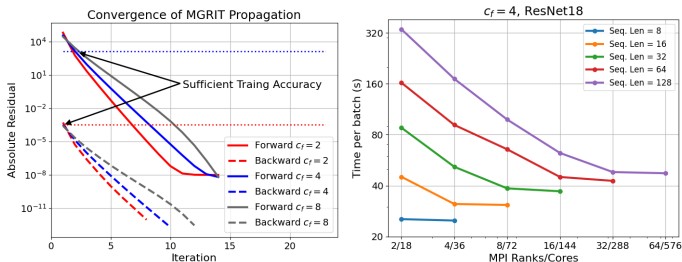

Figure 5: HMDB51 Subset: (left) MGRIT converges rapidly for $c_f = 2$, $4$, and $8$. Sufficient accuracy for training is achieved with $2$ forward iterations, and $1$ backward. (right) Parallel performance of training with different sequence lengths.

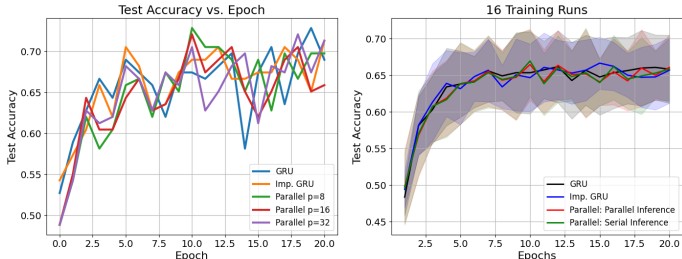

Figure 6: HMDB51 Subset: The accuracy for GRU, and implicit GRU, using serial and parallel training. The right image highlights serial vs. parallel inference with the MGRIT trained network.

## 4.2 HMDB51 DATASET

Fig. 5 presents parameter studies elucidating the scalability and performance of the MGRIT algorithm. The left image in Fig. 5 shows the residual norm as a function of MGRIT iterations for both forward (solid lines) and backward (dashed lines) propagation. The colors differentiate coarsening factors. This demonstrates that the MGRIT algorithm converges in iterations marginally faster for smaller $c_f$. Note that the deeper hierarchies implied by $c_f = 2$ can be more expensive, while $c_f = 8$ may not provide sufficient robustness. As a result, $c_f = 4$ is used in the remainder. Based on the convergence plots and experimentation, only $2$ iterations for forward propagation and $1$ iteration of backward are used for training (sufficient accuracy indicated in the figure). Further, iterations do not improve training; a similar observation was made in Gunther et al. (2020) and Kirby et al. (2020). The right image in Fig. 5 shows the parallel performance over multiple sequence lengths. While there is some degradation in the weak scaling, the run time for a $128$ timestep dataset is roughly only twice that of a dataset with sequences of length $8$. Further, MGRIT performance improves for all lengths if the number of timesteps per MPI rank is $4$ or greater. When timesteps per rank are too few, the computational work is dominated by parallel communication yielding no speedup. Sequences of length $128$ are used in the remaining studies in this section.

Fig. 6 (left) shows test accuracy achieved for a subset of the HMDB51 dataset (see dataset details in Table 2). The Adam optimizer is used with a fixed learning rate of $10^{-3}$. The accuracy for the parallel algorithms are nearby the result obtained for classic GRU. Note that the use of different processor counts does not yield a different algorithm. In the right image, the differences between the schemes are examined after training 16 times. Here, we see that the standard deviation (shaded regions) and mean (solid lines) for the different approaches overlap each other. In addition, both serial and parallel inference of the *Implicit GRU* yield similar results, supporting Remark 3.

To further study the parallel performance of MGRIT training of GRU, we now consider the full HMDB51 dataset. Fig. 7 shows the run time per batch (with a batch size of $100$) when different pre-trained ResNet architectures are used. In all cases, implicit GRU networks are trained with 32 MPI ranks and 9 OpenMP threads (using a total of 288 CPUs), achieving roughly $6.5\times$ speedup over the classic GRU algorithm trained with 9 OpenMP threads. The largest amount of speedup occurs after only 4 MPI ranks. Increasing parallelism to 64 MPI ranks did not result in any speedup. Note

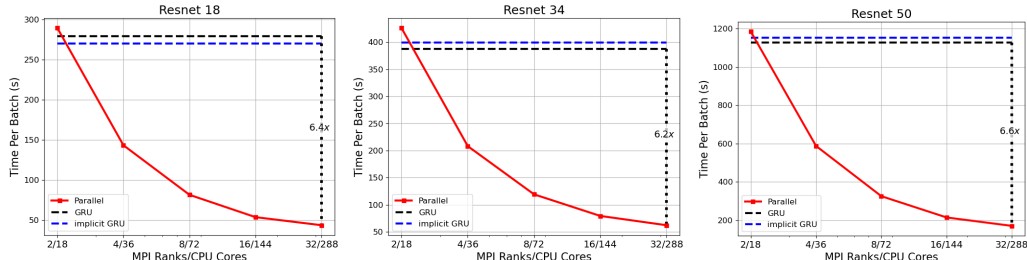

Figure 7: For HMDB51: Batch run times for training a GRU network with ResNet18, 34, and 50.

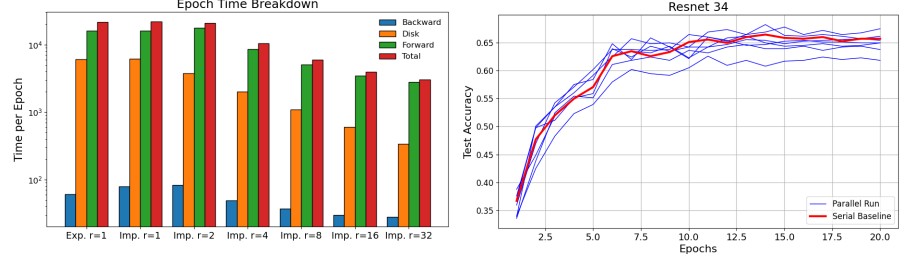

Figure 8: For HMDB51: (left) Per epoch breakdown of run times with ResNet18 preprocessing images. (right) Test accuracy as a function of epochs for a ResNet34 GRU network, trained on 32 MPI ranks with 9 OpenMP threads per rank. The thin blue lines are different initializations for parallel training, and the red line provides a baseline for serial training.

that in this case, each rank is responsible for computing only 2 time steps. Based on Fig. 5 (right), we hypothesize that at this scale the amount of computational work is too small to see benefits from increased parallelism. Fig. 8 (left) shows a breakdown of run times for the ResNet18 case. This *log* plot shows consistent gains with parallelism for the total (red), forward (green), backward (blue) and disk compute times (orange). The expense of the forward simulation is dominated by the evaluation of the ResNet at each time step. The backward propagation cost is reduced because sensitivities with respect to the ResNet are not required. Finally, the disk time can be reduced by simply storing these in memory or overlapping computation and disk read. Note that both of these options are available to the MGRIT training as well. The robustness of MGRIT training applied to implicit GRU using ResNet34 on 32 MPI ranks is investigated with respect to initialization with 7 different random seeds (thin blue lines) and compared to a serial baseline run (red line) in Fig. 8 (right). A learning rate schedule starting from $10^{-3}$ and being reduced by $1/2$ after 5 epochs is used to achieve roughly 65% accuracy for both parallel and serial training. The variation in parallel training is relatively small, and the overall performance is robust. For more details on these studies, including the hyper-parameters, see Appendix A.3.2.

## 5 CONCLUSION

We presented a parallel-in-time approach for forward and backward propagation of a GRU network. Parallelism is achieved by admitting controlled errors in the computation of the hidden states by using the MGRIT algorithm. A key development to make this possible is a new *Implicit GRU* network based on the ODE formulation of the GRU methodology. Applying MGRIT to other types of RNN (LSTM or those proposed in Chang et al. (2019)) is possible with an implicit solver.

The implementation of this approach leverages existing automatic differentiation frameworks for neural networks. PyTorch and its Adam optimizer were used in this work. Comparisons between PyTorch's GRU implementation and the parallel algorithm were made on CPUs using MPI and OpenMP demonstrating a $6.5\times$ speedup for HMDB51 dataset. A GPU version of the MGRIT technology is currently underdevelopment, though related work has demonstrated potential on Neural ODEs (Kirby et al., 2020). Other forms of data and model parallelism are complementary to the proposed technique, though undoubtedly there are unexplored optimizations.

## ACKNOWLEDGMENTS

This work was performed in part, at Sandia National Laboratories, with support from the U.S. Department of Energy, Office of Advanced Scientific Computing Research under the Early Career Research Program. Sandia National Laboratories is a multimission laboratory managed and operated by National Technology & Engineering Solutions of Sandia, LLC, a wholly owned subsidiary of Honeywell International Inc., for the U.S. Department of Energy's National Nuclear Security Administration under contract DE-NA0003525. This paper describes objective technical results and analysis. Any subjective views or opinions that might be expressed in the paper do not necessarily represent the views of the U.S. Department of Energy or the United States Government. SAND Number: SAND2022-0890 C. This work was supported by the National Research Foundation of Korea (NRF) grant funded by the Korea government (MSIT) (No. NRF-2021R1G1A1092597).

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

## A  APPENDIX

### A.1  WHY DOES MGRIT WORK?

To help the reader who may not be familiar with the details of the MGRIT algorithm, we include this brief discussion to provide some intuition about how it works. For brevity, we focus only on MGRIT forward propagation. As discussed above, there are a number of high quality introductory texts on multigrid in general that we recommend for a more in-depth exploration. The monograph "A Multigrid Tutorial" by Briggs et al. (2000) is a really excellent starting place. For MGRIT specifically see (Falgout et al., 2014) and Dobrev et al. (2017). Moreover, a serial implementation of the MGRIT algorithm written in python using `NumPy` is included in the supplementary material (see Section A.4 for a brief overview). The ODE used in this code is

$$\partial_t h(t) = -\frac{1}{2} h(t) + \sigma(Ah(t) + Bd(t) + b). \tag{17}$$

Here, $h$ is the hidden state, $d$ represents the user data, and $A, B, b$ are randomly selected weights and biases independent of time. In the code, the width of the hidden state is a user defined parameter, while in the discussion that follows $h \in \mathbb{R}^3$. This ODE is meant as a simplified representation of an RNN/GRU network. Finally, realize that in the code and in this discussion only forward propagation is considered, no training is performed.

MGRIT is a divide and conquer algorithm where the problem of forward propagation is partitioned into reducing high-frequency error, and low frequency error separately. To this end, the error is written as a decomposition

$$\vec{h} - \vec{h}^* = \vec{e} = \vec{e}_{\text{high-freq}} + \vec{e}_{\text{low-freq}} \tag{18}$$

where $\vec{h}^*$ is the exact solution, $\vec{h}$ is the approximation, $\vec{e}$ is the error, and the remaining two terms are the high-frequency and low-frequency errors. The high-frequency error is reduced in MGRIT by using $FCF$-relaxation. To demonstrate this we consider the ODE in Eq. 17, and choose our initial approximation by selecting random values at each time point to define $\vec{h}$. This seeds the initial error with a number of high-frequency modes. This is evident in Fig. 9 plots (A) and (B), which show the initial error plotted in time and the Fourier coefficients plotted as a function of wave number. Each color represents a different component of the hidden state. The high frequencies are large and significant, even if the low frequency mode is also substantial. Plots (C) and (D) show the error and its Fourier coefficients after 4 sweeps of $FCF$-relaxation. Clearly, from plot (C), the solution has improved in early times in a dramatic fashion. This is expected as $FCF$-relaxation propagates exactly from the initial condition. In addition, the error is reduced in magnitude throughout the time domain. Not so evident from the error plot is the reduction in high frequency errors. This is clearly demonstrated in plot (D), where the low frequency modes remain relatively large while the high frequency modes are substantially damped. Thus, $FCF$-relaxation can be used to reduce high-frequency errors throughout the time domain, and is the first component of the MGRIT divide and conquer strategy.

The second part of the divide and conquer strategy is the coarse grid correction, where a coarse representation of the ODE is used to provide a correction based on the residual from the fine grid. The residual implicitly contains the error in the current guess of the solution on the fine grid. The final pair of images, (E) and (F), shows how the error from the fine grid behaves after it is restricted to the coarse grid, in this case with a coarsening factor of 4. Here, the highest frequency modes from the fine grid are removed from the error, leaving only modes that are low-frequency on the fine grid. The goal of the coarse grid is then to compute a correction to these low frequency modes. If this is the coarsest level, then this computation is a serial forward propagation, which exactly resolves the coarse grid error and the correction is transferred to the fine grid with the prolongation operator. For a multi-level scheme, another round of the divide and conquer algorithm is initiated recursively (applying the MGRIT algorithm again, as discussed in Remark 2). Here, the key is to recognize that parts of the low-frequency error from the fine grid are aliased onto the coarse grid, and become, relative to the coarse grid, high-frequency. This can be seen as a spreading of the Fourier spectrum image (F) relative to image (D). Again, the $FCF$-relaxation algorithm will damp the high-frequency error modes on this level, and the coarse grid correction will correct the low frequency error modes. Regardless, when the recursion returns to the coarse level, the error in both

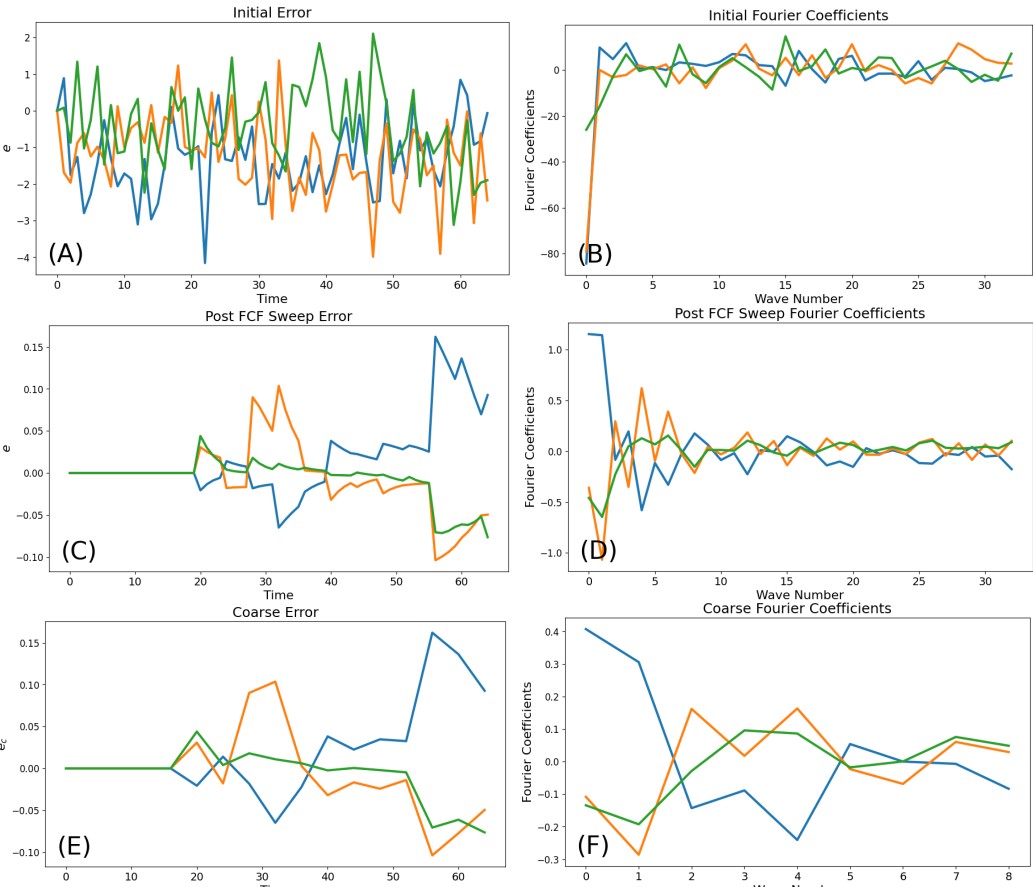

Figure 9: This figure is a sequence of images representing the error in the hidden state for an ODE using $64$ time steps. Each color is a different component of the hidden state (e.g., $h(t) \in \mathbb{R}^3$ for all time). The left column shows the error, while the right column shows the Fourier transform of the error. The first row (images A and B) is the initial error. The second row (images C and D) is the error after $4$ sweeps of $FCF$-relaxation. The final row (images E and F) is the restriction of the relaxed error (in the second row) onto the coarse grid. Note that the scale in the Fourier coefficients, and wave numbers axes vary by row.

the high-frequency modes and low-frequency modes will have been reduced. Thus, a correction transferred to the fine grid will provide further error reduction.

This presentation has focused on forward propagation. However, using an adjoint ODE the motivation and implementation are nearly identical. The most notable difference in the implementation is the reversal of the time evolution.

## A.2 TRAINING WITH MGRIT

Incorporating the time-parallel approach into a standard stochastic gradient descent algorithm requires replacing the forward and backward propagation steps with the equivalent MGRIT computations. Algorithm 3 presents pseudo-code demonstrating how that works. This algorithm requires additional notation for the parameters in the neural network, here denoted by $\Theta$. The structure of the algorithm follows the batched gradient descent approach. In the first part, lines 3-5, the MGRIT algorithm is applied to perform forward propagation. In the second block, lines 6-9, MGRIT is called again, this time to compute the backpropagation through the neural network. Line 10 computes the gradients of the loss with respect to the weights. Note that the sensitivities computed by backpropagation are used to compute the full gradient with respect to the parameters in a parallel fashion. The final line of the algorithm is simply the gradient descent step with a specified learning rate.

---

**Algorithm 3:** MGRITTraining($l_r$, Batches, ForwardIters, BackwardIters)

```
    // This is a training loop over epochs, and mini-batches
1.  for e ∈ 1...Epochs
2.      for each x⃗ ∈ Batches
            // Iterate with MGRIT to perform forward propagation
3.          h⃗ = 0⃗
4.          for i ∈ 1...ForwardIters
5.              h⃗ = MGProp(x⃗, h⃗)
            // Iterate with MGRIT to perform backward propagation
6.          w⃗ = 0⃗
7.          w_T = ∇_{h_T} L
8.          for i ∈ 1...BackwardIters
9.              w⃗ = MGBackProp(x⃗, h⃗, w⃗)
            // Compute the gradient with respect to parameters
10.         g_i = w_i ∂_{Θ_i} h_i    ∀i
11.         Θ⃗ = Θ⃗ − l_r g⃗
```

---

The algorithm specified shows MGRIT embedded within a broader stochastic gradient descent algorithm. However, the code described is already relatively modular. Lines 3-5 are simply forward propagation. Lines 6-10 use back propagation to compute the gradient. In our own `PyTorch` implementation, we were able to encapsulate the forward propagation in a `torch.Module` evaluation call. Similarly, the gradient computation is implemented to be compatible with `PyTorch`'s autograd capability. As a result, utilization of `PyTorch`'s optimizers, like Adam, is straightforward.

A final point needs to be made about the application of MGBackProp in Algorithm 3. This is meant to imply that the MGRIT algorithm is applied in a parallel-in-time fashion to the backpropagation algorithm. In our opinion, the most transparent way to do this is by considering the adjoint of Eq. 15 and its discretization. This is the approach taken in the Neural ODE paper (Chen et al., 2018).

## A.3 COMPUTATIONAL EXPERIMENTS

### A.3.1 UCI-HAR DATASET

The UCI-HAR dataset is used as an initial evaluation of the effectiveness of the MGRIT training for GRU. The parameters used for this study are listed in Table 1. The subsets of labeled *Training Parameters* are values used for both the computational resources and algorithmic details of the optimizers. Note that the number of threads used per MPI Rank is only 1 as this dataset is relatively

| | | |
|---|---|---|
| **Training Parameters** | Optimizer | Adam |
| | Learning Rate | 0.001 |
| | Batch Size | 100 |
| | OpenMP Threads per MPI Rank | 1 |
| **GRU Parameters** | Seq. Length | 128 |
| | Hidden Size | 100 |
| | Num. Layers | 2 |
| | Time step ($\Delta t$) | 1.0 |
| | $T$ | 128.0 |
| **MGRIT Parameters** | Iters Fwd/Bwd | 2/1 |
| | $c_f$ | 4 |
| | Levels | 3 |

Table 1: Parameters used for the UCI-HAR dataset learning problem in Section 4.1.

| | Training Videos | Test Videos | Classes |
|---|---|---|---|
| Full Dataset | 6053 | 673 | all 51 classes |
| Subset of Dataset | 1157 | 129 | chew, eat, jump, run, sit, walk |

Table 2: The full HMDB51 and the subset of HMDB51 are partitioned using a the $90/10$ split between training and test data. The subset contains only 6 of the 51 classes of the original dataset.

small. This implies, on our architecture that contains 36 cores per node, only one node is used for this simulation. Those labeled *GRU Parameters* are network architecture specific parameters. All of these apply to *Classic GRU*, serial *Implicit GRU*, and parallel *Implicit GRU*. Note that a time step of $\Delta t = 1$ is always chosen to ensure alignment with the *Classic GRU* architecture. Finally, the *MGRIT parameters* have been chosen by following the recommendations in prior works of Gunther et al. (2020) and Kirby et al. (2020).

### A.3.2   HMDB51 DATASET

The HMDB51 dataset is composed of 51 classes, and $6726^2$ different videos. This full dataset was used in the generation of the results in Fig. 7 and Fig. 8. The subset of the data was used to generate Fig. 5 and Fig. 6. The breakdown of training to testing videos and the specific classes used are detailed in Table 2. The images in the video are cropped and padded so the image sizes are all $240 \times 240$. Further, to focus on the performance of the MGRIT algorithm, we truncate/pad (with blank frames) the video sequences to a specified length.

Table 3 shows the parameters used for the simulations. These follow the same grouping convention as the parameters for the UCI-HAR dataset. This dataset is substantially larger and more computationally expensive than the UCI-HAR dataset. As a result, more computational resources are used. In the table, this is demonstrated by the use of 9 OpenMP threads per MPI rank. On the platform used, this implies there are 4 MPI ranks per node (each of the 36 cores on a node gets assigned a thread). While we experimented with different configurations, we were unable to get more performance by increasing the number of threads per MPI rank. We suspect that this has to do with the configuration of the memory, and software stack on our system. The table also shows the types of studies and parameter variations we have performed and documented in the present work. Parameter names denoted with an asterisk are varied in our studies. For instance, Adam is used as the optimizer and $\Delta t = 1$ on the fine grid in all experiments, while the learning rate is varied in the experiment producing Fig. 8.

Images from the video sequence are preprocessed during training and inference into reduced features (size 1000, also specifying the GRU hidden size) using pre-trained ResNet18, 34 or 50 networks. In training, the weights and biases of the ResNet's remain fixed. Further, the MGRIT algorithm computes the application of the ResNet only once on the fine grid. The coarse grids make use of

---

[2]This differs from the documented number somewhat. At the time we downloaded these videos, several were unavailable.

| | | |
|---|---|---|
| **Training Parameters** | Optimizer | Adam |
| | Learning Rate* | 0.001 |
| | Batch Size | 100 |
| | OpenMP Threads per MPI Rank | 9 |
| **GRU Parameters** | Seq. Length* | 128 |
| | Hidden Size | 1000 |
| | Num. Layers | 2 |
| | Time step ($\Delta t$) | 1.0 |
| | $T^*$ | 128.0 |
| | Image Size | $240 \times 240$ |
| **MGRIT Parameters** | Iters Fwd/Bwd* | 2/1 |
| | $c_f^*$ | 4 |
| | Levels* | 3 |

Table 3: Baseline parameters used for the HMDB51 dataset learning problem in Section 4.2. These parameters serve as the defaults. However, those that are marked with an asterisk are varied during the study.

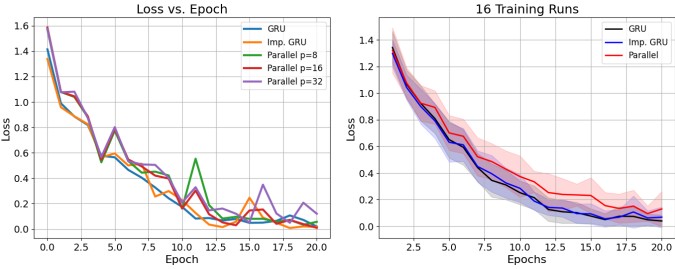

Figure 10: For a Subset of HMDB51: The training loss for the *Classic GRU* method and *Implicit GRU* using the parallel MGRIT propagation algorithm.

those precomputed features to save computational time. In the MGRIT backpropagation step, the features are also reused.

The experiments producing Fig. 5 and Fig. 6 used the smaller subset dataset. For the residual convergence study, the MGRIT coarsening rate was varied. For $c_f = 2$, the number of levels used was 6, with $128, 64, 32, 16, 8$ and $4$ timesteps on each level (from fine to coarse). For $c_f = 4$ (the baseline), the number of levels used was 3, with $128, 32$ and $8$ timesteps on each level. Finally, for $c_f = 8$, the number of levels used was 2, with $128$ and $16$ timesteps on each level. A minimum number of timesteps of $4$ on the coarse level was used that defined the depth of the MGRIT hierarchy. For the performance scaling study in the right image of Fig. 5, varying sequence lengths were used, and, as a result, the time domain parameter $T$ also varied. The remaining parameters followed the baseline from Table 3, notably using $c_f = 4$, and a pre-trained ResNet18.

For Fig. 6, ResNet18 was used and all parameters followed the baseline. In addition to the test accuracy, we include an analogous figure with the loss in Fig. 10. In all cases, the loss decreases steadily regardless of the number of processors. Varying training over multiple initial states yields a similar reduction in loss for the MGRIT algorithm.

Figures 7 and 8 use the full data set. All performance simulations use the baseline parameter set with a sequence length of $128$. The plot of the timing breakdown uses ResNet18. Finally, in the full dataset accuracy results using ResNet34 in Fig. 8 (right), the baseline parameters are used but the learning rate is varied. The learning starts from $10^{-3}$, and then is reduced by $0.5$ every 5 epochs, terminating at $1.25 \times 10^{-4}$.

### A.4 SUPPLEMENTAL MATERIAL: SERIAL MGRIT CODE

To help the reader new to MGRIT, we have included a serial implementation of the two-level algorithm using python with minimal prerequisites (only `NumPy`). It has also been included in our

TorchBraid library as an example and a teaching tool. This example does no training, but produces only one forward solve of the ODE in Eq. 17. By default, the ODE has dimension 10, $\alpha = 1.0$, and the number of steps is 128. The result of running this example is:

```
Two Level MG
  error = 1.6675e-01, residual = 1.1287e-01
  error = 3.1409e-03, residual = 1.9173e-03
  error = 5.4747e-05, residual = 3.9138e-05
  error = 3.5569e-07, residual = 2.6001e-07
  error = 3.2963e-09, residual = 2.9624e-09
  error = 5.5450e-11, residual = 3.6914e-11
  error = 5.7361e-13, residual = 3.9258e-13
  error = 5.1118e-15, residual = 4.5498e-15
```

As the weights and biases are randomly selected, the specific values may be different depending on the seed. However, in general, the error reduction should be similar. The code itself is commented. Further, the implementation of the two level algorithm is labeled with the line numbers used in Algorithm 2.

