# OpenReview forum: "Parallel Training of GRU Networks with a Multi-Grid Solver for Long Sequences"
_ICLR.cc/2022/Conference — ICLR 2022 Poster_

### Official Review · Reviewer_Y8yE · 2021-11-02

**Correctness:** 4
**Technical Novelty And Significance:** 2
**Empirical Novelty And Significance:** Not applicable
**Recommendation:** 6
**Confidence:** 4

**Main Review:**

Major comments:

-) The written English can be improved.

-) The paper is hard to follow as the authors assume that readers are familiar with MGRIT. I understand that this is a technical paper which is geared towards researchers that are seasoned in the topic, and there is also a page limitation. Nonetheless, the authors should put every possible effort to include details, even in the form of an appendix.

-) Unfortunately, it is not clear at all what factors affect parallelism and scalability.

-) Restricting all MPI processes within the same node does not really lead to interesting results. Large-scale networks will need way more resources than a 2-socket Intel node. In fact, I am not sure I understand why there is no benefit beyond 36 MPI processes, at least for the larger examples.

-) MPI is undefined -- please do so and add a proper citation. The appendix should provide guidance as to how the experiments are organized and might be replicated. I understand that this might be a considerable effort but any additional information will be particularly
helpful.

-) The main drawback of this paper is novelty. There is already some work done in the field (the authors cite it properly) and it is not
clear to me what is the main differentiation factor. MGRIT is discussed for example in the paper by Kirby et al. (2020) and I am not
sure that simply focusing on GRUs (together with the discussion in Section 3.3) warrants a publication at ICLR.

-) The main positive part of this paper is the speedup and engineering of making MGRIT work for GRUs. Again, I do not think this is a
major advancement in the field, especially for such a small number of MPI processes.  The authors claim that a GPU version is under
development. This would be a very nice addition -- note though that the related paper cited by the authors uses MPI+GPU with experiments on up to several tens of GPUs.

-) Are hyper-parameters used? I did not see any detailed study. How the algorithm performs for different combinations of T and c_f?
Maybe the authors include some and I missed them.

-) Figures 5, 8, and 9 are too small. Figures 8 and 9 could be combined together.


Minor comments:
Section 4.1:

-)"When training with MGRIT three levels are with a coarsening rate of 4 are used for the length 128 sequences." Please rephrase.
-)"Here we see that the the test accuracy." Remove "the".

**Summary Of The Paper:**

The main focus/goal of the submitted paper is the parallelization of the Gated Recurrent Unit (GRU) (the authors focus on classification problems). The authors describe the incorporation of a multigrid reduction in time (MGRIT) solver to speed-up and better parallelize the
application of forward and back propagation of information. The proposed technique seems to provide a speedup of about an order of
magnitude (at most) when implemented on distributed/shared memory hybrid computing environments.

**Summary Of The Review:**

I find the idea to apply MGRIT to GRUs interesting, however I am not sure that there is sufficient differentiation (and major advancement) between the methodology discussed in this paper and previous work cited by the authors (I am aware of some of this paper by my own reading). This is further pronounced by the fact that previous work also considers larger hardware installations of MGRIT; the amount of
parallel resources considered by the authors does not allow for a trustworthy scaling analysis as inter-node communication is not currently present.

---

> ### Author Response · Authors · 2021-11-17
> **Response to Reviewer 4: Part 1**
>
> First thank you for your comments, they have helped improve the paper. We agree, and apologize for the challenges with the writing. We have made a thorough pass through to improve it. In addition we have added the MPI reference as requested. And taken into account some of your comments about the readability of the figures by increasing font sizes and as suggested combining Figures 8 and 9. Finally, we have addressed your minor comment. At this point we are holding back the revisions so that we can incorporate new results that you and other reviewers have requested. In the mean time, we would like to address some of the other criticisms you have raised.
>
> >  Restricting all MPI processes within the same node does not really lead to interesting results. Large-scale networks will need way more resources than a 2-socket Intel node. In fact, I am not sure I understand why there is no benefit beyond 36 MPI processes, at least for the larger examples.
>
> In terms of the computational resources, we have miscommunicated what we use. We have endeavored to improve the discussion to make it more clear. For the HMDB problem each MPI rank contains 9 OpenMP threads corresponding to 9 CPU cores within a socket (each CPU has 18 cores). Thus only 4 MPI ranks are used per node. To achieve 32 MPI ranks requires 8 nodes, giving (8 nodes)*(4 ranks)*(9 threads) = 288 cores. While not a massive computation from a computational science perspective, it does demonstrate the potential scalability for the approach.
>
> As to why there is no benefit to go beyond 32 MPI ranks, this is simply a matter of running out of work to do
> (e.g. hitting the limits of Amdahl’s law). The longest sequences we ran were 128 images long. When running on 32 MPI ranks that leaves 4 time steps per MPI rank (again each with 9 threads/cores). At this point we didn't see further improvement from going to 64 MPI ranks where each MPI rank is responsible for only 2 time steps. We added the additional text below in Section 4.2. to clarify this point. We thank the reviewer for drawing our attention to this improvement.
>
> “Additionally, increasing parallelism to $64$ MPI ranks did not result in any speedup. Note in this case each rank is responsible for computing only $2$ time steps. We hypothesize that at this scale the amount of computational work is too small to see benefits from increased parallelism.”

---

> ### Author Response · Authors · 2021-11-17
> **Response to Reviewer 4: Part 2a**
>
> > The main drawback of this paper is novelty. There is already some work done in the field (the authors cite it properly) and it is not clear to me what is the main differentiation factor. MGRIT is discussed for example in the paper by Kirby et al. (2020) and I am not sure that simply focusing on GRUs (together with the discussion in Section 3.3) warrants a publication at ICLR.
>
> We believe there are several major contributions of this work that differentiate it from the previous efforts in (Guenther 2020, Kirby 2020). First, the application to recurrent neural networks was not considered in any prior works. Thus this is the first application to RNN’s generically and GRU’s specifically. Note that this is a particularly important domain area for parallel-in-time as the length of the time domain is based on the length of the input data sequence. This is distinct from the Neural ODEs/ResNets where the length of the time domain (number of layers) is dictated by the depth of the network.
>
> Second, the comparison between a parallel approach to inference and a serial one is novel within the cited references.  This is important to recurrent neural networks as their storage costs do not grow with the number of sequence entries. Moreover, a method that trains in a parallel but must be evaluated in parallel would be suspect as errors incurred during training are inherently “learned” as part of the neural network response. In Kirby 2020, we’ve been unable to find a similar study, that work appears to focus primarily on the runtime of forward propagation and inference (please correct if we’ve missed something). Guenther 2020 on the other hand does look at both the computed loss and test accuracy compared to the serial case. However, they consider only the neural network (ResNets in this case) evaluated using the MGRIT algorithm. Thus this aspect, inference in serial and parallel, appears to be entirely novel and a critical addition to prior efforts.
>
> A third novel aspect is the use of a batch gradient descent algorithm as opposed to the one-shot method used in Guenther 2020 and referred to in Kirby 2020. In fact in Kirby 2020, we are having a hard time determining what training algorithm was used for their results (if you are more familiar with the work, it would be helpful to point to the part of the text in the results that discusses this). Thus comparing this to the Guenther 2020 paper, applying a SGD style algorithm is an additional novel aspect.
>
> The final aspect which we view as novel is the development of a stable coarse grid. While in many ways this is contained in the statement “engineering of making MGRIT work for GRUs,” we believe this is an oversimplification of what has been developed here. For instance in Guenther 2020 they acknowledge in the footnote on page 9 that “a more thorough treatment of stability...will be the topic of future research”. In general, choosing a coarse operator (minimally stable) in multigrid methods is challenging business. This is evidenced by the three multigrid for PDEs references below with one in each of the past three decades (Cleary 1998, Falgout, 2014, MacLachlan, 2007). Moreover, while MGRIT leverages an ODE view of neural networks, a general application of multigrid to neural network architecture broadly will require innovation on both the side of relaxation and coarsening. The present effort again leverages the ODE form. However, further investigation into the stability aspects of the ODE are explored. In particular, determining what modes are stiff and need to be handled implicitly is critical if a coarse grid correction is going to be effective. Further, in a deviation from what was done in Guenther 2020 (its not so clear what Kirby 2020 uses for the time step parameter ‘h’) we used a uniform time step of $\Delta t =1$ (which we admit is not clearly stated). This implies that the approach we used is close to the original GRU, though the noted implicit time step is necessary in order to maintain stability on the coarse grid. This is important because it limits the modification to the GRU network and doesn’t require a smaller time step that might change the behavior. A final point to be made here, is that this exploration provides another data point for the potential of MGRIT. However, if a naive application of GRU was explored  without the adjustments for stability, then its poor performance may be improperly attributed to the MGRIT algorithm.
>
> Cleary, A. J., R. D. Falgout, V.E. Henson, and J. E. Jones. "Coarse-grid selection for parallel algebraic multigrid." In International Symposium on Solving Irregularly Structured Problems in Parallel, Springer, Berlin, Heidelberg, 1998.
>
> Falgout, R. D., and J. B. Schroder. "Non-Galerkin coarse grids for algebraic multigrid." SIAM J. on Scientific Computing 36, no. 3 (2014): C309-C334.
>
> MacLachlan, S., and Y. Saad. "A greedy strategy for coarse-grid selection." SIAM J. on Scientific Computing 29, no. 5 (2007): 1825-1853.

---

> ### Author Response · Authors · 2021-11-17
> **Response to Reviewer 4: Part 2b**
>
> ...continuation of 2a:
>
> Based on these points we believe the present work more than meets the novelty threshold. Moreover, applying multigrid to a new application area or neural network will most often require innovation like that presented in section 3.3. Multigrid training is in its infancy and its potential for success in accelerating training and inference will require a series of advances that will improve its robustness and breadth of applicability. We believe, building on Guenther 2020, and Kirby 2020, that the present work is such an advance. We’ve added the following text to the introduction to further clarify the novel contribution:
>
> “However, this training algorithm will not lead to accurate networks due to stability problems on the coarse grid. This emphasizes the challenges of choosing a proper coarse grid model for neural networks, and multigrid algorithms in general. To alleviate this we develop a new GRU architecture, or discretization of the ODE, which we refer to as \emph{Implicit GRU} that handles stiffness in the architecture. This is required for application of the MGRIT algorithm where sub-sequences providing the correction, correspond to discretizations with more severe numerical stability restrictions. An additional novelty of the present work compared to prior MGRIT efforts is the use of Adam with mini-batches to train. Finally, we explore the accuracy of inference when done in serial versus parallel. This ensures that the network is not compensating for the error introduced by the MGRIT procedure, and is a study that had not been done before.”

---

### Official Review · Reviewer_ix54 · 2021-11-02

**Correctness:** 4
**Technical Novelty And Significance:** 3
**Empirical Novelty And Significance:** 2
**Recommendation:** 6
**Confidence:** 4

**Main Review:**

The paper describes an iterative technique for evaluating GRU networks based on the multigrid reduction in time (MGRIT) technique. Key idea is partitioning of a long sequence into shorter sub sequences. These subsequences are trained in parallel, error corrections between grids are applied to correct for inexact computation.

The paper is well-organized, and the mathematical descriptions of the method are clear. The application of the method to GRU layers is novel and will likely be a useful strategy for overcoming the inherently sequential nature of the GRU. The authors describe the tradeoffs in play here well: more resources will be effectively utilized (to a limit noted in the discussion) in applying a fast and scalable, though inexact, method at the cost of a small amount of accuracy in the trained model relative to exact evaluation of the GRU. Obviously, a GPU implementation of this technique would be interesting as well, and the authors note that this is in development. Even without it, though, the results are compelling enough.

Related work is incomplete, recent work in sequence modeling are particularly missing.

More details on “Training with MGRIT” (section 3.4) are needed. In the absence of a reproducible (anonymized) source code, pseudo code or code snippet would provide more clarity (sufficient as supplementary materials).

More could be done in experimental evaluations e.g., quantify impacts of coarsening factor on model quality.

There were some minor grammatical issues; careful copy-editing is recommended. Figure 6 is missing legend.

**Summary Of The Paper:**

The paper describes a technique for evaluating GRU networks based on the multigrid reduction in time (MGRIT) technique. These techniques are not new, in general or to neural network training, but the contribution here is their application to GRU layers. After presenting some of the theory behind ordinary differential equation (ODE) representations of the GRU and laying out the mathematical framework for the MGRIT method, the method is evaluated on two datasets. The results show impressive scalability up to 32 processes (CPU-only) at the cost of a slight loss of accuracy compared to the traditional, sequential GRU

**Summary Of The Review:**

A paper describing a novel parallelization strategy for GRU networks that adds dramatic scalability with some minimal loss in accuracy.

---

> ### Author Response · Authors · 2021-11-17
> **Response to reviewer 3**
>
> Thank you for the constructive feedback and specific suggestions.
>
> We have made a number of changes to the text based on your comments about grammatical issues. We apologize if these problems degraded readability. We have also included some additional references to sequence modeling in the related work discussion in the introduction. Below we comment on each of your points.
>
> > More details on “Training with MGRIT” (section 3.4) are needed. In the absence of a reproducible (anonymized) source code, pseudo code or code snippet would provide more clarity (sufficient as supplementary materials).
>
> We have added a matching “Training with MGRIT” section to the appendix that goes into more detail. The idea here is to explore the application of MGRIT in the context of SGD. This requires a variant of the MGRIT algorithm applied to the adjoint of the GRU-ODE, which is similar to what is done in the Neural ODE paper (Chen et al, 2018). We also highlight how the MGRIT procedure can be made compatible with existing neural network frameworks, in our case PyTorch. These can be embedded both as a forward evaluation “Module” and to compute gradients within the autograd tool suite.
>
> > More could be done in experimental evaluations e.g., quantify impacts of coarsening factor on model quality.
>
> We are working to add further experimental evaluations. The first is focused on considering how performance depends on the length of the sequence. The second evaluation, based on your comment, will consider how the coarsening factor impacts training and accuracy. In this context the coarsening factor is somewhat orthogonal to accuracy. The MGRIT algorithm can always be iterated further. It is interesting though to understand that for small coarsening factors (e.g. 2) the computational cost will be higher per iteration. For larger coarsening factors (e.g. 8) the cost per MGRIT iteration will be lower, but it may require more iterations to get to the same accuracy. In this sense, the question you ask is extremely pertinent because its speaks to why we chose a particular coarsening factor for our experiments. As a result we will include results in the revision that speak to this.

---

### Official Review · Reviewer_2GTo · 2021-11-03

**Correctness:** 3
**Technical Novelty And Significance:** 3
**Empirical Novelty And Significance:** 3
**Recommendation:** 6
**Confidence:** 3

**Main Review:**

Strengths:
In this paper, authors propose a novel parallel-in-time (PinT) training method based on a MGRIT solver for GRU networks with long sequences. This method has been well motivated by illustrating the limitations and unsolved problems of the existing methods.
The proposed algorithm enables the accelerated parallel training of GRUs on long sequence, which is a unique capability that permits growth in this dimension.
Experimental results provided on the HMDB51 dataset is impressive, which demonstrate the proposed parallel training scheme of GRU achieves up to 6.5× speedup over a serial approach.
Overall, the paper is well written and clear, especially the part of methodology introduction.

Weaknesses:
Can you give a discussion about the different performance of proposed method in parallel inference versus serial inference?
There is a confusion that regarding the test accuracy, the experiment setting for UCI-HAR and HMDB51 dataset is a bit different. Can you show the results/plot in the same manner or provide an explanation to it?
Can you provide the test accuracy of baseline in Figure 9 as well? Otherwise, the accuracy performance of MGRIT for full HMDB51 is unsure, which is necessary to demonstrate whether the remarkable speedup on HMDB51 dataset is meaningful.
The experiment results are not clear to demonstrate the statement in Remark 3, “Our results below indicate consistency between serial and parallel inference even when training is done with MGRIT. ” Consequently, confusion is introduced for the statement as follows, “This suggests that error introduced by MGRIT is not a severe limitation.” A more convincing illustration is expected.

Comments:
Grammar errors(underline): Sec. 4.1 - “When training with MGRIT three levels are with a coarsening rate of 4 are used for the length 128 sequences. ”
In Sec. 4.1 - “Here we see that the the test accuracy using parallel inference (red line) is modestly lower then purely serial case. ”, “then” -- > “than”
Grammar errors(underline): Sec. 4 – “In this section, we evaluate the performance our new parallel-in-time training approach on the sequence classification tasks such as human activity recognition and video classification. ”
In Conclusion – using “6.5\times” would be better than “6.5x(letter x)”

**Summary Of The Paper:**

This paper aims to address the limitations of existing approaches for training Gated Recurrent Unit (GRU) given long sequence in terms of both training time and model accuracy. To tackle this challenge, author propose a novel parallel training scheme (called parallel-in-time) for GRU based on a multigrid reduction in time (MGRIT) solver.  Specifically, the key to achieving speedup is a hierarchical correction of the hidden state to accelerate end-to-end communication in both the forward and backward propagation. Authors gives experimental results on two public datasets to demonstrate the performance improvement in the long sequence scenario.

**Summary Of The Review:**

N/A

---

> ### Author Response · Authors · 2021-11-20
> **Response to Reviewer 2**
>
> Thank you for your comment on this. Looking back through our data in fact we did compute the serial performance for the result in Figure 6 (HMDB51 subset data). And it does align well with the result for the UCI-HAR data set. The original purpose of this plot was to demonstrate how the performance was independent of processor count. However, we feel based on your comments that a better story can be told by discussing the serial inference in more detail instead. Thus, to save space the loss portion of Figure 6 will be moved to the appendix. We are working on a new set of results (currently running) that will mirror that of the UCI-HAR simulation, but on the HMDB51 subset data. And should provide more evidence for the correctness of the serial inference even when trained with MGRIT. We believe that satisfies your critique about Remark 3. The results show that the serial inference parallel inference are well aligned with the serial Classic GRU (better even the the UCI-HAR data set).
>
> As to the test accuracy base for Figure 9. We are trying to run this result as we would agree it would be valuable in strengthening the argument. It’s not clear we will be able to have the result in time for the deadline of 11/22 (we have been hampered by system level disk issues, and general challenges of a queuing system). However, we believe that the preponderance of evidence in the UCI-HAR data set, and the HMDB51 data set (with the new results showing inference on the subset) is sufficient to conclude the algorithm can successfully train on even large data sets. Bear in mind, we have demonstrated both the variation in the full data set using MGRIT and the parallel scalability used to get there. Hopefully, you will agree.

---

### Official Review · Reviewer_bzSV · 2021-11-05

**Correctness:** 4
**Technical Novelty And Significance:** 3
**Empirical Novelty And Significance:** 3
**Recommendation:** 8
**Confidence:** 2

**Main Review:**

First of all, I have to admit that I am not very familiar with the multigrid reduction in time solver technique and its application to ordinary differential equations. My understanding of this paper is thus not very deep.

Strengths:
1. The paper provides a new view to parallelize the training task of machine learning models. Previously, progresses in parallelization techniques have lend great power to the efficiency of parallel training, and helped reduce idle time when many devices are used to train large models with large amount of training data. Each new idea in this domain is a great step forward.
2. The paper is based on solid techniques (multigrid reduction in time solver) that have been used in other domains (solving ordinary differential equations). Pivoting a proven technique to new applications always lends more credibility to the proposed solution.


Weakness:
1. Since the parallelization technique in this paper gives best performance for recurrent neural networks with long sequences, the impact of this paper might not be that wide.
2. The GRU architecture has to be adapted to achieve the result mentioned in the evaluation. This is a bit limiting because machine learning professional might have their reason to use a specific recurrent architecture, or might want

Questions:
Since I am not very familiar with the details of the technique, I would rather use this reviewing opportunity to ask a few questions.
1. If my understanding is not way-off, the technique trades off computation for parallelization. In other words, more computation is needed for the same number of iterations of training, so that the GRU can be parallelized at the `time` dimension. If this understanding is correct, I am curious where exactly the speed up (6.5 times) is from. Is it because we can utilize the multiple devices more efficiently (there is less idle time on the devices)? Could I say that for the same number of iterations of training, more FLOPs are actually needed, but less time is taken since the devices are running in parallel? If the same number of iterations of training is ran, should we expect the technique to have the same degree of model accuracy?

2. Let's assume that the model is as simple as a 2 layer GRU. If we unroll the model, it just looks like a long repetitive model with lots of GEMMs and activations. There are still data dependencies between the repetitive units, just like the GRU data dependencies. For the unrolled model, there are published parallelization techniques (such as GPipe and PipeDream). I think GPipe might require the batch size to be large enough for good parallelization factors, but PipeDream seems to be very good even if the batch size is not big. PipeDream also has certain inaccuracy, in the sense that the gradients are always stale. I am just wondering how do the authors compare this technique with PipeDream?

Nits:
1. in page 2, the second usage of ODE has explanation "ordinary differential equation", but it should be explained in the first use (end of line 2)
2. in page 4 (around the middle) there is a "the the hidden state".
3. in page 6 section 3.4, there is a "fewer then would be required" but I think it should be "fewer than would be required"



**Summary Of The Paper:**

The paper proposed to parallelize the inference and training of GRU networks (a type of recurrent neural networks) at the `time` dimension.
The main contribution is the application of multigrid reduction in time (MGRIT) solver, and a new GRU architecture (Implicit GRU) that handles the stiffness in the architecture. As a result, the evaluation shows 6.5 times faster training time.

**Summary Of The Review:**

I think the paper provides a very interesting new approach of `time-parallelization`, which can benefit the machine learning community a lot. Thus I think this paper can be accepted.

---

> ### Author Response · Authors · 2021-11-16
> **Response to Reviewer 1**
>
> Thank you for your questions.
>
> - Response to question 1
>
> We will answer your first question in reverse order. The algorithm we employ is an iterative one. At each step of gradient descent an MGRIT solve is used to compute the forward and backward solve inexactly. To exactly perform the forward/backward substitution requires iteration to fixed convergence (a residual value of zero). **If the algorithm is allowed to converge to zero, then we would expect the same model accuracy as when training with serial forward/backward propagation.** In this case we are solving the same problem using many more FLOPS than in the serial case (as you suggested).
>
> If the iteration is stopped short of complete convergence, then the forward and backward propagation is done inexactly. But, that inexact solution is computed in a parallel fashion. Again, this will likely require more FLOPS, as even after one iteration of MGRIT the amount of work required exceeds one serial gradient calculation. However, **the amount of work per iteration, in FLOPS, required by a single processor run in parallel is much less than then the work required by computing a gradient on a single processor in serial. Thus, the total runtime per iteration is far less in parallel than in serial.** The utilization of the processors in parallel is somewhat inefficient. If we have 32 way parallelism we would hope to have 32$\times$ speedup. But, because each iteration is globally more FLOPs than a step of gradient descent in serial we don’t achieve ideal speedups and our parallel efficiencies are degraded. However, we still achieve real speedups (6.5$\times$) despite this issue. It’s an open, and interesting, question how one can achieve greater parallel efficiencies.
>
> Your original question is where the parallel speedups come from. What enables parallel speedups is the inexact propagation through the network. Each iteration, while performing more FLOPS globally, requires fewer FLOPS on each processor yielding a reduced runtime assuming the iteration count is small. Thus, figure 5 is critically important. This figure shows the error reduction per iteration for a forward and backward solve. Experimentally we’ve determined, and it’s inline with previous literature (Gunther 2020, Kirby 2020), that relatively few iterations are needed for gradient descent. Again, this is an interesting research topic that we will explore in future work.
>
> - Response to question 2
>
> As you mentioned, if we unroll the GRU model, the parallelization approach proposed by PipeDream might be possible for parallelizing the GRU model across time dimension.
> Let’s assume that we have 4 available processors (workers), the number of mini-batches is 50, and the length of each sequence in a mini-batch is 128.
> Given these settings, as we can imagine from Figure 4 in the PipeDream paper (refer to a published version, not an arXiv version of the paper), PipeDream splits an entire sequence of size 128 into 4 sub-sequences of size 32. Then each processor maintains a pipeline to take multiple mini-batches of a set of sub-sequences. Each processor then computes the forward pass for multiple mini-batches included in its pipeline in serial manner.
> In their pipelining approach, the output of the first mini-batch 1 computed from the processor 1 has to be transferred to the processor 2 as an input to the mini-batch 1 on processor 2. This implies that processor 2 must wait until processor 1 finishes the forward computation for the mini-batch 1.
> Hence each processor has to wait for (processor id-1)$\times$(time required for processing the first mini-batch) to start running.
> So **the pipelining approach is somewhat performance-limited as the number of processors increases**. Using the pipelining method, it is challenging to reduce the idle time before the start of every processor.
> PipeDream paper described a round-robin based work scheduling scheme for the better load balancing (refer to Figure 8 in their paper), but it still has unpreventable idle time.
>
> In our new parallel-in-time (PinT) approach, the data partitioning over all processors seems to be similar to the PipeDream approach but our parallelization approach is completely different from the pipelining method.
> **The main difference is we don’t use a pipeline for parallelizing the GRU given a sequence of input data.
> All processors immediately start running at the same time and process their mini-batches without any dependency across different processors.**
> After processing multiple mini-batches on a fine grid in each processor, the main worker (processor 1) will only gather the final outputs computed from all the processors to perform coarse grid correction which only requires a small amount of time to process few time steps (e.g., computing only 4(=number of processors used) input data (time steps) of sequence for each mini-batch).
>
> - Thank you for pointing out all the typos. We have corrected the errors in the revised paper.

---

### Author Response · Authors · 2021-11-20
**Overview of Revisions**

We must thank the reviewers for their thoughtful critique. We believe that this has been a positive guide for our efforts to improve on the original document. In what follows we overview some of the larger changes arising from reviewer feedback. Note that we have provided more detailed feedback for each review in response to individual critiques. Finally, in the revision we have included a substantial appendix and supplemental material (source code). Major changes in the text have been highlighted in blue where they correspond to reviewer feedback, while more minor changes for readability and length are not highlighted.

We highlight several broad critiques and then provide a response and point to changes made in the manuscript as a result.

**General Critique**: Lack of familiarity with the MGRIT methodology.

**Resulting Changes**:
We have added an appendix section titled “Why does MGRIT work?” that describes at a high-level some of the intuition behind multigrid. The focus is a classical (geometric) presentation of the algorithm as providing a divide and conquer approach on the frequency space of the hidden state.

A serial implementation of MGRIT has also been added to the supplemental materials that facilitates reader exploration with the MGRIT algorithm.

**General Critique**: The details of applying MGRIT to training are not clear

**Resulting Changes**:
We have added a section in the appendix that presents pseudo-code for the training algorithm. We also briefly discuss how it can be made to fit into a deep learning software frameworks. A high-level description of our PyTorch implementation is included.

**General Critique**: The parallel/serial inference discussion is lacking for the HMDB51 dataset.

**Resulting Changes**:
We have performed more calculations and presented the results of parallel vs. serial inference for a subset of the HMDB51 data set (see Figure 6 (right), and Figure 10 (right) in the appendix). The results indicate agreement between classic GRU, implicit GRU with serial and parallel inference. We believe that this result provides the required support for Remark 3.

**General Critique**: More details about the experiments and computing platforms are needed.

**Resulting Changes**:
We added tables and discussion in the appendix to meet this need. We also strengthened the description of the computational platform in the text. In particular, we highlighted the use of multiple threads per MPI rank, and changed the figures to show the number of cores and ranks used.

**General Critique**: The novelty of the work doesn’t warrant publication.

**Resulting Changes**:
We have highlighted the critical contributions made by this paper. To repeat, the application of MGRIT to GRU’s specifically and RNN’s in general is new. We also developed a new GRU architecture that handles the stability constraints on the coarse grid. Finally, the comparison between parallel and serial inference is novel, and a critical contribution of the present work.

While we believe our work is sufficient novelty, we would also like to point out that the breadth of applicability of MGRIT (and multigrid) to neural network training has not yet been established. This is evidenced by the mixed familiarity of the reviewers on this particular document. The reason this is critical is that multigrid has been an important tool to achieving good parallel scalability in a number of other fields. Thus multigrid maybe an important tool for dramatically accelerating neural network training. We believe this effort, expanding on prior work, is a step towards establishing the broader applicability.

---

### Decision · Program_Chairs · 2022-01-20

**Decision:**

Accept (Poster)

**Comment:**

This paper presents a way of using multigrid techniques to parallelize GRU networks across the time dimension. Reviewers are uniformly in favor of accepting the paper.  The main strength is that the paper provides a new perspective on dealing with long input sequences by parallelizing RNNs across time. The main weaknesses are around the experiments: only CPU experiments are run, and sequences are not very long (max 128 length). All-in-all, though, it provides an interesting perspective that should be valuable to the community.